# Diagnosis and Treatment in Unilateral Condylar Hyperplasia

**DOI:** 10.3390/jcm12031017

**Published:** 2023-01-28

**Authors:** Jorge Beltran, Carlos Zaror, María Paz Moya, Henrique Duque Netto, Sergio Olate

**Affiliations:** 1Department of Oral and Maxillofacial Surgery, Hospital Clínico Regional Guillermo Grant Benavente, Concepción 4070038, Chile; 2Division of Oral and Maxillofacial Surgery, University of Concepción, Concepción 4030000, Chile; 3Center for Research in Epidemiology, Economics and Oral Public Health (CIEESPO), Faculty of Dentistry, Universidad de La Frontera, Temuco 4780000, Chile; 4Faculty of Dentistry, Universidad San Sebastián, Puerto Montt 5480000, Chile; 5PhD Program in Morphological Sciences, Universidad de La Frontera, Temuco 4780000, Chile; 6Faculty of Health Sciences, Universidad Autonoma de Chile, Temuco 4780000, Chile; 7Department of Oral and Maxillofacial Surgery, Federal University of Juiz de Fora, Juiz de Fora 36000-000, Brazil; 8Division of Oral, Facial and Maxillofacial Surgery, Universidad de La Frontera, Temuco 4780000, Chile; 9Center of Excellence in Morphological and Surgical Studies (CEMyQ), Universidad de La Frontera, Temuco 4780000, Chile

**Keywords:** condylar hyperplasia, facial asymmetry, orthognathic surgery and TMJ

## Abstract

Unilateral condylar hyperplasia (UCH) is an uncommon disease involving progressive facial asymmetry. The aim of this research was to perform an analysis of the diagnosis and treatment of patients with UCH in a clinical series. An observational retrospective study was performed on subjects with progressive facial asymmetry in the lower third of the face; all the subjects were under treatment with condylectomy and orthodontics to improve occlusion and face balance. Variables such as age, sex, clinical type, SPECT (single photon emission computed tomography) intensity and a requirement for secondary surgery were included; the Shapiro Wilk test was performed to analyze the normality of the data and nonparametric analysis and the Kruskal-Wallis or Mann-Whitney tests were used to assess the association between the SPECT difference and the variables, where 2-tailed *p* values < 0.05 were considered to be statistically significant. Forty-nine patients between 10 and 45 y.o. (average age: 19.1 ± 7.4 y.o.) were included in the study. There were 41 female (83.6%) and 8 male (16.4%) subjects. The SPECT analysis comparing the right and left condyles with more than 10% in caption of the isotope was present in 46 subjects; the results obtained using SPECT were not statistically related to the age or sex of the sample (*p* = 0.277). The patients were classified into clinical types I, II and III, and no correlations could be confirmed between the clinical type and other variables. High condylectomy was conducted on all patients, among which 14 patients underwent a secondary surgery for orthognathic or cosmetic treatment, and was not related to the initial variables used in diagnosis (*p* = 0.98); interestingly, the second surgical treatment was more present in the clinical type I and in subjects under 16 years old with no statistical differences. Clinical analysis, medical records, 3D imaging and SPECT should be used as a complementary analysis in assessing the diagnosis of UCH and progressive facial asymmetry.

## 1. Introduction

Unilateral condylar hyperplasia (UCH) is a rare disease involving the middle and lower third of the face; UCH was described by Adams in 1936 and diagnosis remains controversial, although some protocols have been reported for the diagnosis of progressive facial asymmetry [1]. Clinical photographs, 2D or 3D images, SPECT and dental casts have been proposed as diagnostic routes [2,3] (Figure 1).

Some reports have shown that single photon emission computed tomography (SPECT) can be effectively used in diagnosis to identify abnormal growth in a mandibular condyle [4]; however, other reports have shown poor specificity and sensitivity of SPECT as a diagnostic test, with 30% to 60% concordance [5].

In terms of clinical classification, UCH was first proposed by Obwegesser and Makek [6] and later by Wolford et al. [7], showing characteristics of the disease as a type of deformity, age and clinical considerations about treatment (Figure 2 and Figure 3).

Age has been considered an important variable to understand the evolution of facial asymmetry; in this sense, early treatment could resolve facial asymmetry and dental occlusion in better conditions than later treatment [8] (Figure 3).

SPECT and clinical characteristics are included in the diagnosis of UCH (Table 1); however, those variables have not been clearly correlated with the diagnosis and progression of facial asymmetry. The aim of this research was to perform an analysis of the diagnosis and treatment of patients with UCH in a clinical series. 

## 2. Material and Methods

Observational retrospective research in consecutive patients was designed. Included in this research were 49 male and female subjects, consulted for progressive facial asymmetry, with an age between 10 and 45 years old. Patients with a history of facial trauma, congenital facial deformity or treatment of benign or malignant maxillofacial tumors were excluded. All patients approved the study protocol and signed an informed consent form; this study was conducted in accordance with the guidelines of the Helsinki Declaration and approved by the Ethical Committee for Research with the Protocol UFJF-IRB 2.148.583.

All the subjects were submitted to clinical analysis. The progression of the facial asymmetry in the last year was confirmed subjectively by patients and family; subjects were grouped by facial asymmetry related to clinical type and dental occlusion [6]. Subsequently, a CBCT (MORITA, Veraviewepocs 3D R100, resolution images of 125 μm voxel, field of view (FOV or area exposures) Ø 40 × H 80 mm) of the temporomandibular joint was obtained, and SPECT (Sophy DST-Xli model, open-gantry SPECT gamma camera, SMV, France) was performed to compare the right and left sides of the face using the OSEM 3D (Ordered Subsets Expectation Maximization; Siemens Medical Solutions, Erlangen, Germany) reconstruction method with 3D collimator beam modeling and optional attenuation. SPECT, as a quantitative variable, was used to serve as an independent variable.

The data included in the analysis were sex, age, clinical type, surgery and a SPECT report. The clinical type was classified as type I, type II and type III [6]. Age was analyzed to find differences between the younger and older groups; the SPECT analysis was used to determine differences between the left and right condyles related to the caption of the radioisotope (Table 2).

After diagnosis, all the patients were treated using condylectomy of the hyperplastic condyle and orthodontics; orthognathic or cosmetic surgery was performed a second time, when necessary, to achieve facial symmetry, dental and facial balance and normal function.

All the samples with a UCH diagnosis were included in the statistical analysis. The Shapiro Wilk test was performed to analyze the normality of the data. Given the clearly skewed distribution of the SPECT difference, we decided to use nonparametric analysis, and Kruskal-Wallis or Mann-Whitney tests were used to assess the association between the SPECT difference and covariables. Two-tailed *p* values <0.05 were considered statistically significant. The data analyses were performed using Stata 15 (Stata Corp, College Station, TX, USA).

## 3. Results

Forty-one female (83.6%) and 8 male subjects (16.4%) were included with an age between 10 to 45 years old (19.1 ± 7.4 years old). The left side was affected in 27 cases (55.2%) and the right side was affected in 22 cases (44.8%). 

In all subjects, the UCH and progressive asymmetry in the lower third of the face was confirmed; SPECT images obtained using ^99c^Tc showed differences between the right and left sides, showing 11 patients with 10% or less in difference. Specifically, the difference between the left and right condyles was 6% in 3 cases, 10% in 8 cases and between 12% and 48% in the remaining 36 cases (Table 1). Sex was not related to the SPECT results and the intensity of the SPECT was not associated to the female or male (*p* = 0.174).

The clinical types were classified as type I for 22 subjects (44.9%), type II for 10 cases (20.4%) and type III for 17 cases (34.7%) and was not correlated with the SPECT results (*p* = 0.058). 

The average age was 16.8 (3.5) for clinical type I patients, 22.7 (11.2) for clinical type II patients and 20.6 (5.6) for clinical type III patients; although a lower average age and a low standard deviation were found for clinical type I group when compared to type II and type III, no statistical relation was observed between age and clinical type (*p* = 0.354). 

High condylectomy of the mandibular condyle and orthodontics were performed in all patients to treat UCH. A second surgical time with orthognathic surgery or cosmetic surgery to treat residual facial asymmetry was performed on 14 subjects (28.5%) with no relations to any variable revised in the initial diagnosis as SPECT results, the age in the first surgery or any other variable (*p* = 0.98). In the clinical type I, only one subject (4.5%) was involved in a secondary surgery (Table 2); in the clinical type II, 5 subjects (50%) were included in a secondary surgery (Table 3) and in the clinical type III, 8 subjects (36.7%) were treated with secondary surgery to get better function or esthetic conditions (Table 4). No statistical differences were observed (*p* = 0.443) between groups. 

Interestingly, subjects with diagnosis and treatment realized between 10 and 16 years of age had only 2 subjects (9.5%) require a second surgical procedure (Table 5 and Table 6); after 17 years of age, the unilateral high mandibular condylotomy was followed by a second surgical treatment in 12 subjects (42.8%). Even with this difference, no statistical differences were observed in age related to second surgery requirements (*p* = 0.189).

## 4. Discussion

Unilateral condylar hyperplasia (UCH) is a rare disease related to the lower third of the face, leading to facial asymmetry [9]. The reason for consultation is usually progressive facial asymmetry and changes in dental occlusion [10]. Incidence of this pathology is difficult to obtain because of the uncommon nature of the disease.

Female predominance in UCH was reported previously in some articles [1,10] showing a clear trend. In this research, no statistical differences were observed; however, 83.6% of subjects were female and only 16.5% were male. It has been discussed that the hormonal conditions in the female could be involved in the development of UCH; however, there is no consensus in this field [3,4,5]. Most likely, a statistical difference between male and female could be observed in a larger sample. 

Clinical presentation usually shows chin deviation and asymmetric class III dental occlusion because the nonaffected condyle grows normally or close to normally and the affected side grows abnormally large [3,7,10], moving the neck, ramus and body of the mandible in a forward and contralateral direction, resulting in chin deviation and progressive facial asymmetry (Figure 4).

In the series of 49 patients included in this study, progressive facial asymmetry was observed as the main reason for consultation, and all patients showed chin deviation and asymmetry in dental occlusion.

UCH is diagnosed by clinical approach; 2D-rx is not effective for diagnosis [11] because it does not show the size and morphology of the condyle, the ramus and body of the mandible. 3D imaging, such as computed tomography (CT) or cone beam computed tomography (CBCT), shows the volumetric condition and size of the condyle, and can be used to compare the right and left condyles [12,13]; 3D imaging can be used to observe the cortical line of the condyle and to assess the bone quality as an indicator of bone metabolism.

All 49 patients presented an augmented condyle in the affected side, showing hyperplastic growth of the condylar head. Goulart et al. [12,14] reported that the bilateral hyperplastic condyle has the same size, morphology and volume as the condyle involved in UCH, demonstrating that a class III dental occlusion could be related to bilateral hyperplastic condyles, as previously proposed by Wolford [6] and Obwegeser [7].

The main use of SPECT in UCH treatment is to perform a comparison between the condyles. In 1985, Matteson [15] demonstrated the utility of ^99m^Tc in bone scanning to assess abnormal condylar growth, and in 2007, Lippold et al. [16] determined the biological basis for using a bone scan to assess the activity level of the condyle. SPECT analysis was used for diagnosis, and no correlations were observed between age, sex, clinical type or the uptake of ^99m^Tc and the SPECT results. Karssemakers et al. [13] investigated 20 subjects with an average age of 22.8 y. (for an age range of 10 to 39 y.o.) and did not find any correlation between the trabecular bone volume fraction and trabecular thickness of the condyle and the condylar activity on the preoperative bone scan, showing that the bone volume was not related to bone activity [17].

In histological analysis, Fariña et al. [18] found no correlation between histological conditions and SPECT results, and Saridin [19] found low consistency in the correlation used to evaluate the uptake of ^99m^Tc and the histologic evidence. A SPECT analysis cannot be used to distinguish among inflammatory, infective or healing processes [20]. Some reports [5] show difficulties in obtaining accurate results when comparing different techniques for SPECT measurements in UCH diagnosis because the considerable variation in terms of sensitivity and specificity. SPECT measurement can be related not only to abnormal condylar growth, but to bone blood flow, vascular permeability and bone metabolism [21,22]. 

Lopez et al. showed that SPECT was not related to age or sex; they also showed that the specificity of the SPECT could be improved using clinical and tomographic study to quantify the mandibular deviation [23]. Age could be a significant variable in some characteristics of the active UCH. Saridin [24] showed variability in the cartilage thickness and islands in resected condyles and that age was inversely related to the thickness of the cartilage layer. Previous histological results from our group led to similar conclusions [25,26]. Nitzan et al. [27] reported a large clinical series showing a larger impact of UCH in patients under 30 years old, and Wolford’s classification was related to age in the initial stage of UCH. For that reason, SPECT must be used in the armamentarium for UCH diagnosis, with other variables as shown in this research. 

UCH was present in 21 subjects between 11 and 16 y.o. (42.9%) and 28 subjects between 17 and 45 y.o. (57.1%), showing a higher incidence of the disease in the initial 6 years of adolescence than during adulthood. A low metabolism and the absence of a growth process of the condyle in the cases of older subjects could be the main reason for this difference [13]. 

In terms of the clinical type, type I (horizontal) was present in 45% of subjects with an average age of 16.8 y. (±3.5), type II (vertical) was present in 20% of subjects with an average age of 22.7 y. (±11.2) and type III was present in 35% of subjects with an average age of 19.9 y. (±7.8); no statistical relationship between the clinical type and the age was observed. Villanueva-Alcojol [3] recorded the clinical distribution of the facial type, showing 66.7% of patients were type I with an average age of 21.7 y. o. (±7.4), 22% of patients were type II with an average age of 25.6 y.o. (±3.2) and 11.1% of patients were type III with an average age of 26 y.o. (±7.1). Clinical type I would be common in younger patients, but no statistical relations were observed.

Between 10 and 15 years old, the condylar head can grow in a vertical position over 10 mm in females and 15 mm in male patients, which is related to facial bone growth [28] under normal and abnormal conditions. Faster movement of the facial bone is observed in younger patients because of the adolescent growth process. Usually, the neck, ramus and body of the mandible grow proportionally with the condyle, so a large condyle indicates a large mandible, and a short condyle indicates a short mandible [29,30]; this fact has been confirmed by the results in an animal model showing mandibular modification based on the use of growth factors in the condyle [31]. 

However, as facial growth has stopped in adults, different characteristics for the condyle are observed in adults vs. adolescents; this difference could be an argument for the differences of clinical behavior in the UCH and the SPECT variations regarding specificity and sensibility. Indeed, Shetty and Guddadararangiah [32] showed a sequence of pictures of a patient from early age to adult age, showing great differences in terms of facial morphology related to progressive facial asymmetry.

High condylectomy was used to treat the affected side in all the patients. Fourteen (28.5%) patients were submitted to a secondary surgical treatment for orthognathic or cosmetic surgery; this secondary treatment was not related to the variables included in the diagnosis.

Interestingly, in the case of second surgical treatment and after high condylectomy, this was present in 4.5% of the clinical type I, in 50% of the clinical type II and in 36.7% of the clinical type III. No statistical differences were observed; however, it is possible to assume that more complex deformities are present in the clinical type II and III and this can be the reason for more treatment requirements.

In the same line, second surgical treatment was observed in 9.5% of subjects under 16 years old and in 42.8% of the subjects between 17 and 45 years old; while there were no observed statistical differences, it could be assumed that the early treatment can help in a better resolution of the functional and esthetic concerns.

Limitations in this research are related to the observational nature of the investigation. The absence of groups to compare the variables and the results of treatment with could be a problem to define a better protocol for the diagnosis and treatment of subjects with UCH. 

No independent variables included in this research show exclusive correlations with UCH diagnosis and treatment. Clinical analysis, medical records, 3D imaging and SPECT should be used as a complementary analysis to assess the diagnosis of UCH and progressive facial asymmetry.

## Figures and Tables

**Figure 1 jcm-12-01017-f001:**
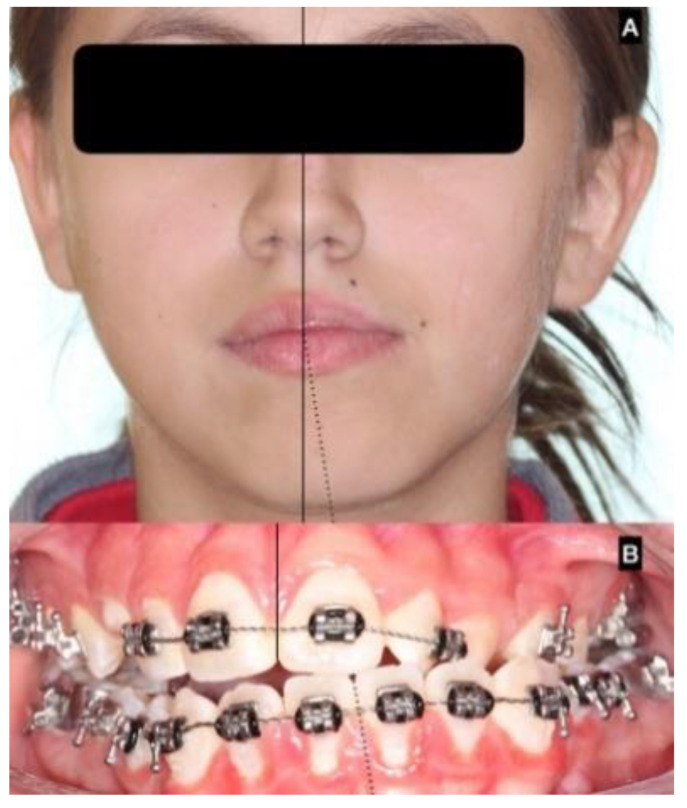
Clinical presentation of unilateral condylar hyperplasia and facial asymmetry. (**A**) chin deviation with a class III trend; (**B**) lack in dental midline and unilateral crossbite dental occlusion.

**Figure 2 jcm-12-01017-f002:**
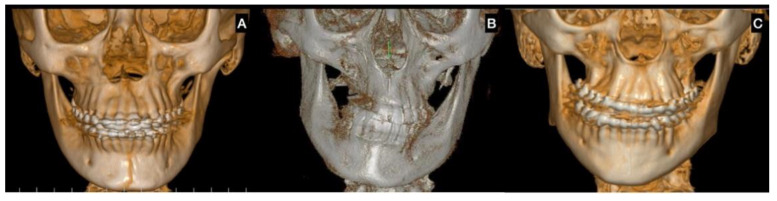
Classification of the facial asymmetry related to unilateral condylar hyperplasia. (**A**) type 1, with a horizontal component with UCH on the right side; (**B**) type 2, with a vertical component with UCH on the left side; (**C**) type 3, with a horizontal and vertical component and UCH on the left side.

**Figure 3 jcm-12-01017-f003:**
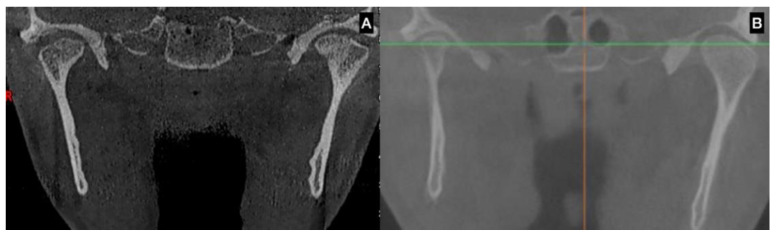
Hyperplastic growth of the condyle on the left side in different subjects showing progressive facial asymmetry. (**A**) Young subject with an augmented condyle on the left side (note the augmented articular space in the right TMJ with a normal size of the condyle). (**B**) Augmented condyle on the left side with more differences in height and width compared to the right condyle in an older subject.

**Figure 4 jcm-12-01017-f004:**
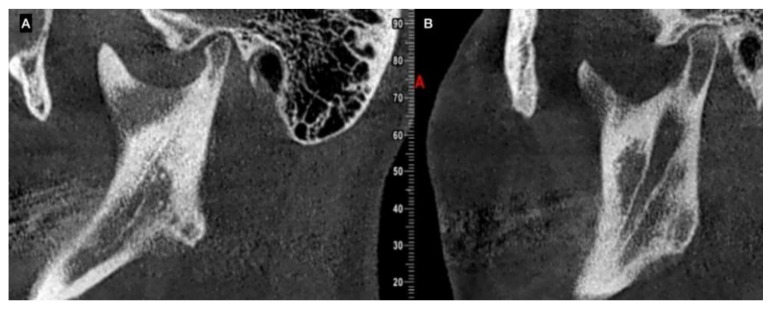
CBCT of the same subject in sagittal view. (**A**) left condyle in normal growth and (**B**) right condyle with hyperplastic growth involving neck and ramus of the same side of the mandible. The progressive asymmetry will move all the sides involved in the abnormal growth of the condyle.

**Table 1 jcm-12-01017-t001:** Variables and criteria related to diagnosis in unilateral condylar hyperplasia applied to this research.

Test	Characteristics and Definition
Family medical record	Family or patient describes a progressive facial asymmetry, occurring in the last time
	With or without any family with facial deformity or facial asymmetry
	Pain or noise in the affected condyle could be reported
Facial analysis	Chin midline deviation with the facial midline
	Asymmetry in mandibular angles (vertical or horizontal differences)
Dental conditions	Can be present with unilateral crossbite in the canine and/or molar area or unilateral open bite mainly in the posterior area
	Progressive deviation of the dental midline with more than 4 mm.
	Trend to class III dental occlusion
CBCT	Augmentation in condylar size in comparison to the non-hyperplastic condyle
	Lack in upper cortical line of the affected condyle in the upper area showing an active metabolism
	Augmentation in radiolucency in the affected condyle with an image related to poor density in some cases
SPECT	Differences 10% in caption of the radioisotope between the hyperplastic and the non-hyperplastic condyle

**Table 2 jcm-12-01017-t002:** Distribution of subjects with UCH according to clinical type I.

Sex	Age	Affected Side	SPECT Differences	Secondary Surgery
F	17	R	12	NO
F	19	L	6	YES
F	14	L	10	NO
F	22	L	16	NO
F	17	L	10	NO
M	22	L	14	NO
F	22	R	10	NO
F	14	L	18	NO
F	22	L	6	NO
F	12	R	48	NO
F	15	R	10	NO
F	14	L	14	NO
F	17	L	22	NO
F	14	L	10	NO
F	17	L	14	NO
F	22	R	18	NO
M	15	R	12	NO
F	10	L	12	NO
M	16	L	26	NO
F	18	R	24	NO
F	13	R	18	NO
M	18	R	16	NO
Average	16.81		15.72	
Standard Deviation	3.55		8.92	

F: Female; M: Male; R: Right; L: Left.

**Table 3 jcm-12-01017-t003:** Distribution of subjects with UCH according to clinical type II.

Sex	Age	Affected Side	SPECT Differences	Secondary Surgery
F	15	R	16	NO
F	23	R	18	YES
F	21	L	16	YES
F	13	L	22	NO
F	45	L	28	YES
F	14	L	26	NO
F	41	L	10	YES
F	16	L	26	YES
F	20	R	24	NO
M	19	L	20	NO
Average	22.7		20.6	
Standard Deviation	11.2		5.66	

F: Female; M: Male; R: Right; L: Left.

**Table 4 jcm-12-01017-t004:** Distribution of subjects with UCH according to clinical type III.

Sex	Age	Affected Side	SPECT Differences	Secondary Surgery
F	18	L	10	YES
F	14	L	16	NO
F	21	L	10	NO
F	16	R	20	YES
F	22	L	16	YES
F	14	L	22	NO
F	22	R	23	YES
F	18	R	24	YES
F	35	R	10	YES
F	14	R	26	NO
F	16	R	40	NO
F	17	R	14	YES
F	14	R	12	NO
F	16	R	12	NO
F	17	R	16	NO
M	22	L	28	NO
M	43	L	22	YES
Average	19.94		18.88	
Standard Deviation	7.84		7.99	

F: Female; M: Male; R: Right; L: Left.

**Table 5 jcm-12-01017-t005:** Distribution of subjects with UCH between 10 and 16 years old.

Sex	Age	Affected Side	SPECT (R/L)	SPECT Differences	Clinical Type	Secondary Surgery
F	10	L	44/56	12	1	NO
F	12	R	74/26	48	1	NO
F	13	L	39/61	22	2	NO
F	13	R	59/41	18	1	NO
F	14	L	42/58	16	3	NO
F	14	L	45/55	10	1	NO
F	14	L	41/59	18	1	NO
M	14	L	39/61	22	3	NO
F	14	L	43/57	14	1	NO
F	14	L	37/63	26	2	NO
F	14	L	45/55	10	1	NO
F	14	R	63/37	26	3	NO
F	14	R	56/44	12	3	NO
F	15	R	58/42	16	2	NO
F	15	R	55/45	10	1	NO
M	15	R	56/44	12	1	NO
F	16	R	60/40	20	3	YES
F	16	R	30/70	40	3	NO
F	16	L	37/63	26	2	YES
M	16	L	37/63	26	1	NO
F	16	R	56/44	12	3	NO
Average	14.23			19.8		
Standard Deviation	1.48			9.9		

F: Female; M: Male; R: Right; L: Left.

**Table 6 jcm-12-01017-t006:** Distribution of variables of subjects with UCH between 17 and 45 years old.

Sex	Age	Affected Side	SPECT (R/L)	SPECT Differences	Clinical Type	Secondary Surgery
F	17	R	56/44	12	1	NO
F	17	L	55/45	10	1	NO
F	17	L	39/61	22	1	NO
F	17	R	57/43	14	3	YES
F	17	L	43/57	14	1	NO
F	17	R	58/42	16	3	NO
F	18	L	45/55	10	3	YES
F	18	R	62/38	24	3	YES
F	18	R	62/38	24	1	NO
M	18	R	58/42	16	1	NO
F	19	L	47/53	6	1	YES
M	19	L	40/60	20	2	NO
F	20	R	62/38	24	2	NO
F	21	L	45/55	10	3	NO
F	21	L	58/42	16	2	YES
F	22	L	42/58	16	1	NO
M	22	L	43/57	14	1	NO
F	22	R	55/45	10	1	NO
F	22	L	42/58	16	3	YES
F	22	L	47/53	6	1	NO
F	22	R	61/39	23	3	YES
F	22	R	59/41	18	1	NO
M	22	L	36/64	28	3	NO
F	23	R	59/41	18	2	YES
F	35	R	55/45	10	3	YES
F	41	L	45/55	10	2	YES
M	43	L	39/61	22	3	YES
F	45	L	36/64	28	2	YES
Average	22.75			16.32		
Standard Deviation	7.98			6.28		

F: Female; M: Male; R: Right; L: Left.

## Data Availability

The data are available upon request from the corresponding author.

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
