# Peer review of "Diagnosis and Treatment in Unilateral Condylar Hyperplasia"

_jcm, 2023, doi:10.3390/jcm12031017_

Round 1
Reviewer 1 Report (Previous Reviewer 3)
Revisions in article shows insufficient input on UCH:
- 3rd title change
- content lacks diligence
- poor grammar
- non contributary to UCH literature
Author Response
Dear reviewer
Thank you for your opinion. We do not know if you rejected the article or asked to change something into the text.
This manuscript was revised by some reviewers and they asked to modify some points and were attended by the authors.
We think that this manuscript shows a good analysis about a large case series of a rare disease. We included interesting data to improve the knowledge about the diagnosis and treatment of UCH. The data ​are​ clear and the tables are helping other authors for more analysis as in the case of a systematic review. We think that this document helps to know about the progress in treatment, follow up of subjects submitted to condylectomy and the requirements for more surgery and finally the relation between complementary exams.
Reviewer 2 Report (New Reviewer)
Dear Authors,
could you please try to explain the different prevalence in males and females?
Abstract Explain the acronymous SPECT
Solve occlusion ... improve occlusion
Add the aim of the study.
Please revise the paper to focus the aim of the study.
Remove the double numbering in the references.
Author Response
Rev: could you please try to explain the different prevalence in males and females?
Reply: this paragraph was modified in the discussion about the suggested topic: Female predominance in UCH was reported previously in some articles [1,10] showing a clear trend. In this research, no statistical differences was observed, however 83.6% was female and only 16.5% was male. Has been discussed that the hormonal conditions in the female could be involved in the develop of UCH, however there is no consensus in this field [3-5]. Probably, in a larger sample could be observed a statistical difference between male and female
Rev: Abstract Explain the acronymous SPECT
Reply: thank you, explained in the section
Rev: Solve occlusion ... improve occlusion
Reply: thank you, was modified
Rev: Add the aim of the study.
Reply: the aim was added in the abstract and in the main text to be I accordance with the manuscript.
Rev: Remove the double numbering in the references.
Reply: thank you. References were modified
Reviewer 3 Report (New Reviewer)
The authors investigated cbct reviews and treatment status of uch patients. However, I have minor concerns.
· First of all, I think that the introduction section is insufficient and should be sliglty expanded.
· It is necessary to add the total number of patients to the material section.
· The authors stated that they divided the patients into type 1-2-3. What they mean should be stated or referenced in the material section.
· I think the discussion section is enough.
Author Response
Rev: First of all, I think that the introduction section is insufficient and should be sliglty expanded.
Reply: dear reviewer, thank you for your input. We designed an introduction in this way because the early reviewers asked us to modify this point; so, this text was a result of the initial requirement for publication.
In the same direction and with this introduction, we created a strong discussion chapter including a lot of elements, data and scientific background to confirm our reflections about the UCH. If you agree, we can maintain the format of this introduction with some changes realized into this chapter.
Rev: It is necessary to add the total number of patients to the material section.
Reply: the number of patients was included in the M & M
Rev: The authors stated that they divided the patients into type 1-2-3. What they mean should be stated or referenced in the material section.
Reply: was modified this paragraph in M & M: The clinical type was classified as Type I, Type II, and Type III [6];
Rev: I think the discussion section is enough.
Reply: Thank you
This manuscript is a resubmission of an earlier submission. The following is a list of the peer review reports and author responses from that submission.
Round 1
Reviewer 1 Report
Review
The abstract should be rewritten with separate headings for aims and objectives, materials and methods, results and conclusion.
Aims and objectives of the study.
Not mentioned
Materials and methods
The study states that it is retrospective observational study. How did the participants then approve the study if it was retrospective? It is not clear if the study is retrospective or prospective.
Results
Please elaborate the below statement.
Sex was not related to the SPECT difference (p=0.174).
Conclusion
Is the article concluding that SPECT is of limited utility in the diagnosis and management of Unilateral Condylar Hyperplasia?
Kindly explain the below statement.
“No independent correlations of the variables included in this research were exclusively related to UCH diagnosis. Measurement in SPECT was no related to the UCH. Clinical analysis, medical records, 3D imaging and SPECT should be used as a complementary analysis to assess the diagnosis of UCH and progressive facial asymmetry.”
Kindly consider framing this particular statement in a better way so that the meaning is clear. The language used is ambiguous and does not make for an easy read. The sentences are overtly long and the language used is grammatically incorrect. These are the few examples.
“Clinical classification of UCH was first performed by Obwegesser and Makek [6] and later by Wolford et al. [7], showing characteristics of the disease; age has been considered a key factor to understand the evolution of the facial asymmetry; in this sense, the late treatment of facial asymmetry related to UCH could not be effective when compared to early treatment [8].”
Age was not related to the SPECT intensity (p=0.277); was grouped the subjects be-tween 10 and 16 y.o. because the entire body is actively growing during these ages and compared the SPECT results with those of the group with ages between 17 and 45 yr; although a higher SPECT intensity was found for the younger group, no statistical relationship between age and the SPECT intensity was found.”
“In this study, condylectomy was used to treat the affected TMJ in all the patients, and 14 (28.5%) patients were under secondary surgical treatment for orthognatic surgery or cosmetic surgery; this requirement was not related to the initial SPECT analysis or any other variable.”
Is the following abbreviation acceptable ‘16 yr.’?
.
Only one reference is from 2020 onwards
The authors can add these references to make the reference list more contemporary
1) López DF, Ríos Borrás V, Muñoz JM, Cardenas-Perilla R, Almeida LE. SPECT/CT Correlation in the Diagnosis of Unilateral Condilar Hyperplasia. Diagnostics (Basel). 2021;11(3):477. doi: 10.3390/diagnostics11030477
2) Shetty S and Guddadararangiah S. Case Report: unilateral condylar hyperplasia [version 1; peer review: 2 approved]. F1000Research 2021,10:46
(https://doi.org/10.12688/f1000research.48499.1)
Reviewer 2 Report
Review of „Variables Related to Diagnosis of Unilateral Condylar Hyperplasia”
This paper aims to describe the relation of variables like age, sex, clinical type, intensity of SPECT and requirement of secondary surgery to the underlying diagnosis of unilateral condylar hyperplasia UCH). Although an interesting subject, there are certain questions I have for the authors:
1. INTRODUCTION
a. As the diagnosis of UCH is somewhat controversial, in my opinion it should be clearer what the diagnosis consists of. Those might be summed up in a table with noting which parameters the authors agree on and which they do not (e.g. progressive facial asymmetry, SPECT analysis) to sharpen the scope of the paper in identifying the relevant variables.
b. Also, incidences in the general population would help to get an impression of the uncommonness of the disease.
c. It would be helpful to have a picture of a UCH patient as a visual correlation of the disease.
d. The clinical types of unilateral condylar hyperplasia could be shown as CBCT scans for the same reason, and the description of the differences that has been given in the discussion should be moved to the materials and.methods section.
e. The aim of this study is not really clear to me. As it is depicted in the results section, all variables were correlated to SPECT analysis and not to the diagnosis of UCH. This needs to be clarified.
2. MATERIALS AND METHODS
a. Was the patient group selected by consecutive patients? Was the age group inclusion determined based on previous papers or was it only a normal distribution of presenting patients? A clarification is needed.
3. RESULTS
a. From the position of the p value it is not clear if the p values are correlated to the overall patients in the line or just the line the are put in, e.g. the middle line of the clinical type etc. It would be better to have an overall n for the overall p value in addition to the subgroup p values.
b. Also, there is no legend for the table. No explanation is given to n, x, SD, especially x needs clarification.
4. DISCUSSION
a. What do the authors make of the strong female-leaning sex distribution of the patient group? Is this backed by incidences or just a result of patient selection? Clarification would be great.
b. What do the authors think are the weaknesses of the paper? Normally, in low incidence diseases the case number may be a problem, but the authors have gathered a large study group that would strengthen the relevance of the paper. A paragraph in the discussion would be useful.
c. After the investigation of the correlation of variables to SPECT, do the authors think it is a necessary and/or needed investigation for the diagnosis of UCH? DO they still use it in their practice? What are the suggested consequences of the work for the further treatment algorithm of UCH? A paragraph and a conclusion addressing these points would be beneficial for the clarity of the paper.
The authors have collected a fair amount of cases with good diagnostic data of UCH. However, the scope of the paper should be defined more precisely (maybe even in the title), as it focuses on the relevance of SPECT imaging in the diagnosis and treatment of UCH. As SPECT is an expensive and also psychologically straining examination, the relevance in the treatment algorithm should be determined and a suggestion given based on this data.
The paper is interesting to read, but needs some more polishing and structure. The references are good, and the time frame of 1985-2020 just reflects the fact that we are dealing with a rare disease while the majority has been published in the last decade. I hope that with my comments I could contribute to make this paper better and suitable for publication after the changes.
SUMMARY:
Novelty: The work should be better defined in terms of scope to provide an advancement of the current knowledge, i. e. do we really need SPECT for the diagnosis of UCH?
Scope: With corrections, the work could fit the scope of JCM.
Significance: The results need further interpretation and conclusions should be clearer based on the findings. The hypothesis should also be sharper.
Quality: The presentation needs improvement with figures, table modification and clarification of results.
Scientific soundness: Data seems robust enough, but the correlations are to general and not clear enough in terms of variables to SPECT and not to overall diagnosis.
Interest to the readers: The topic may be of wider interest to the readers, because facial asymmetry is very common, and an understanding of rare diagnoses helps to sharpen the treatment algorithms according to the underlying diseases.
Overall merit: If the authors can suggest that SPECT is not quintessential for the diagnosis due to low sensitivity and specificity, the treatment algorithm can be tightened and additional costs for unnecessary investigations can be avoided.
English level: The language in this paper is appropriate and understandable.
OVERALL RECOMMENDATION: Reconsider after major revision
Reviewer 3 Report
Missing characteristics of the study.
Insufficient/ incorrect mention of diagnostic methods and treatment options in UCH. Also study specific.
Numbers don't add up in 2 sections.
discrepancy in abstract and results.
Aim study not clear enough.
(Post) condylectomy protocol unclear.
Role of SPECT scan in study is unclear (variable or not?).
Bold statements concerning SPECT as well as UCH diagnosis.
Incorrect statement concerning timing of treatment
Editing of the english language is appropriate.
Round 2
Reviewer 2 Report
I think the paper is now more polished and can be interesting for readers outside of the general realm of oral and maxillofacial surgery, as it now outlines the etiology of UCH better. The pictures and tables help even more. Also, the focus of the primary goal of investigation on the role of SPECT makes it more concise. I would like to thank the authors for incorporating the suggested changes.